# The ubiquitous flavonoid quercetin is an atypical KCNQ potassium channel activator

Kaitlyn E. Redford[1] & Geoffrey W. Abbott [1✉]

Many commonly consumed plants are used as folk medicines, often with unclear molecular mechanisms. Recent studies uncovered the ubiquitous and influential KCNQ family of voltage-gated potassium (Kv) channels as a therapeutic target for several medicinal plant compounds. Capers - immature flower buds of *Capparis spinosa* - have been consumed for food and medicinal purposes for millennia. Here, we show that caper extract hyperpolarizes cells expressing KCNQ1 or KCNQ2/3 Kv channels. Capers are the richest known natural source of quercetin, the most consumed dietary flavonoid. Quercetin potentiated KCNQ1/KCNE1, KCNQ2/3 and KCNQ4 currents but, unusually, not KCNQ5. Strikingly, quercetin augmented both activation and inactivation of KCNQ1, via a unique KCNQ activation mechanism involving sites atop the voltage sensor and in the pore. The findings uncover a novel potential molecular basis for therapeutic effects of quercetin-rich foods and a new chemical space for atypical modes of KCNQ channel modulation.

---

[1] Bioelectricity Laboratory, Department of Physiology and Biophysics, School of Medicine, University of California, Irvine, CA, USA. ✉email: abbottg@uci.edu

  1

Human species have been using plants for medicinal purposes since prehistoric times[1–3]. Medicinal use of many of the same plants consumed by our ancestors for their purported therapeutic effects has persisted to the present day[4]. Evidence for their efficacy is in some cases supported by clinical trials but is often purely anecdotal[5,6]. In addition to preclinical and clinical evidence, an understanding of the molecular mechanisms underlying the actions of plant extracts and the compounds within them can inform our understanding of modern and ancient folk medicines. In addition, some plants utilized as folk medicines are consumed globally as food by billions of people each year, in many cases on a daily or weekly basis. Therefore, a greater understanding of the molecular actions of the components of such plants is warranted.

We and others recently discovered that membrane proteins within the KCNQ (Kv7) gene subfamily are particularly responsive to certain compounds produced by plants, including those used for folk medicine and food[7–10]. Each of the five KCNQ genes (KCNQ1 through KCNQ5) encodes a voltage-gated potassium (Kv) channel α subunit. Four such α subunits combine to form a functional Kv channel, with each α subunit comprising 6 transmembrane segments (S1-S6). S1-4 form the voltage sensing domain (VSD); S5-6 make up the pore module (Fig. 1a). Kv channels are activated by cell membrane depolarization, sensed by the S4 within the VSD, which contains periodic basic residues. S4 movement signals the pore to open, allowing $K^+$ diffusion down its electrochemical gradient —predominantly outward under physiological conditions. This hyperpolarizes the membrane, reducing action potential frequency and/or repolarizing the cell following action potential firing[11].

KCNQ α subunits are each capable of forming functional homomeric Kv channels and in some cases can form heteromeric

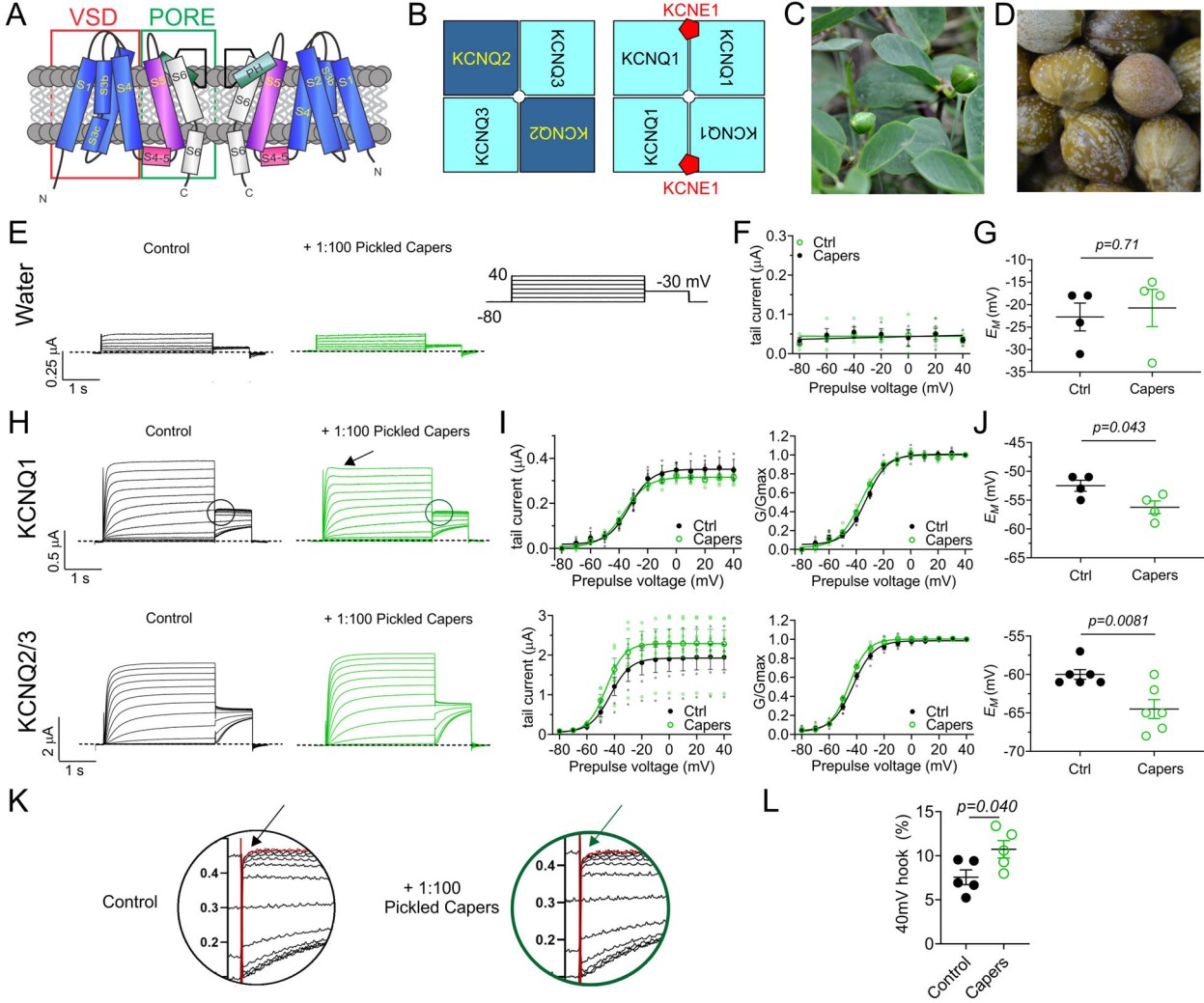

**Fig. 1 Caper extract KCNQ-dependently hyperpolarizes cells.** All error bars indicate SEM. *n* number of oocytes. **a** Topological representation of a Kv channel showing two of the four subunits that comprise a channel. PH, pore helix. VSD, voltage sensing domain. **b** Schematic of heteromeric composition of KCNQ2/KCNQ3 (left) and KCNQ1/KCNE1 (right) channels. **c** Image of caper bush (*Capparis spinosa*). **d** Image of pickled capers used in this study. **e** Mean TEVC current traces for water-injected *Xenopus* oocytes as indicated, in the absence (Control) or presence of 1% caper extract (*n* = 4). Dashed line here and throughout indicates the zero-current level. Right inset, the voltage protocol used here and throughout the study (used with either 10 or 20 mV prepulse increments) unless otherwise indicated. **f** Mean tail current versus prepulse voltage, for traces as in **e** (*n* = 4). **g** Scatter plot of unclamped membrane potential ($E_M$) for cells as in **e** (*n* = 4). Statistical analyses by two-way ANOVA. **h** Mean TEVC current traces for KCNQ1 and KCNQ2/KCNQ3 in the absence (Control) or presence of 1% caper extract (*n* = 4-6). Arrow, current decay. Circles, regions used for close-up images in **k**. **i** Left, mean tail current; right, mean normalized tail current (G/Gmax) verses prepulse voltage for traces as in **h** (*n* = 4-6). **j** Scatter plot of unclamped membrane potential ($E_M$) for cells as in **h** (*n* = 4-6). Statistical analyses by two-way ANOVA. **k** Close-up images of tail hooks (arrows) taken from circled regions in **h**. **l** Quantification of relative size of tail hook to overall peak tail current at +40 mV, from traces as in **h**, *n* = 5.

channels with one another—strong evidence for which exists for KCNQ2/3, KCNQ3/5 and KCNQ4/5 channels. In addition, KCNQ1 channels are highly modulated in vivo by members of the KCNE family of regulatory β subunits, which each bear a single membrane-spanning segment (Fig. 1b). KCNQ2/3, and to a lesser extent KCNQ3/5 and possibly homomeric KCNQ2, KCNQ3 and KCNQ5 channels generate the neuronal M-current, a subthreshold Kv current that regulates neuronal excitability[12–16]. KCNQ4/5 and perhaps homomeric KCNQ1, KCNQ4 and KCNQ5 channels, in some cases regulated by β subunits such as KCNE4, are expressed in vascular smooth muscle and regulate vascular tone[17]. KCNQ1 is also expressed in human heart, the inner ear and a variety of epithelia; in each of these tissues it associates with one or more KCNE isoforms to facilitate the tissue-specific roles it serves[18]. KCNQ4 channels perform roles in auditory neurons and hair cells required for auditory function[19].

In addition to their responsiveness to changes in membrane potential, KCNQ channels are responsive to small molecules, making them partially "ligand-gated". This typically takes the form of a negative shift in the voltage dependence of their activation, such that when the small molecule binds the channel can open at more negative membrane potentials. Thus, direct binding of the neurotransmitter GABA to KCNQ3 or KCNQ5 results in increased channel activity at subthreshold potentials, reducing excitability of cells expressing these channels[20]. Several compounds within plants bind to a similar site to GABA and can induce a similar shift in voltage dependence of activation, which we and others have demonstrated to provide a plausible molecular mechanism for the therapeutic effects of these plants[7–10].

Here, we discovered that 1% extract of capers, which are the immature flower buds of the caper bush (*Capparis spinosa*), KCNQ-dependently hyperpolarizes cells. We describe the molecular basis for this effect, which involves KCNQ activation by a ubiquitous flavonoid, quercetin, via an atypical mechanism and binding site and discuss implications for medicinal and food use of quercetin and the foods that contain it.

## Results

**Caper extract KCNQ-dependently hyperpolarizes cells**. We conducted a methanol extraction (80% methanol/20% water) on pickled capers (*Capparis spinosa*) rinsed to remove excess salt (Fig. 1c, d). Following evaporation of the methanol to leave an aqueous solution of caper extract, we diluted the extract 1/100 in recording solution and screened for effects on non-injected *Xenopus laevis* oocytes versus oocytes expressing KCNQ1 or KCNQ2/3, using two-electrode voltage clamp (TEVC) electrophysiology.

Caper extract had no effect on endogenous current in water-injected control oocytes (Fig. 1e, f) nor their unclamped resting membrane potential ($E_M$) (Fig. 1g). In contrast, the caper extract exerted subtle effects on KCNQ1 and KCNQ2/3 currents, noticeable as an increase in inactivation of the former and current magnitude of the latter (Fig. 1h, i). More markedly, caper extract hyperpolarized the resting membrane potential ($E_M$) of oocytes expressing KCNQ1 or KCNQ2/3 (Fig. 1j). We investigated the apparent increase in KCNQ1 activation in response to caper extract by quantifying the 'hook' at the start of the tail current (circled in Fig. 1h; close-ups in Fig. 1k). The hook is generated by channels passing through an open state on the way from the inactivated state to the closed state during recovery from inactivation prompted by a drop to more negative membrane potentials[21,22]. Caper extract increased the relative size of the hook by 45%, consistent with an increase in KCNQ1 inactivation (Fig. 1l). As the hyperpolarizing effects of caper extract (Fig. 1j) are indicative of increased KCNQ channel activation at

subthreshold potentials, an activity of potential therapeutic importance, we investigated the underlying mechanistic basis, focusing first on KCNQ2/3 channels.

**Caper component quercetin negative-shifts the voltage dependence of KCNQ2/3 activation**. Capers are the richest known natural source of quercetin (Fig. 2a), which is considered to be the most common bioflavonoid (plant pigment) in the plants regularly consumed by humans[23]. Another flavonoid, rutin (also named quercetin-3-O-rutinoside, rutoside, sophorin, or 3,3′,4′,5,7-pentahydroxyflavone-3-rhamnoglucoside) (Fig. 2b) is found in capers at even higher concentrations than quercetin[24]. We found that rutin (100 μM) had negligible effects on KCNQ2/3 activity ($P = 0.324$, $n = 4$–7, at $+40$ mV) (Fig. 2c, d) with no change in voltage dependence (Fig. 2d). In contrast, quercetin (100 μM) negative-shifted the voltage dependence of KCNQ2/3 activation (by $-10.3 \pm 3.4$ mV) without increasing peak currents at positive voltages (Fig. 2e, f). The peak effect of quercetin was a >4-fold current increase at $-60$ mV (Fig. 2g). Consistent with these findings, while rutin had no effect (Fig. 2h), quercetin (100 μM) hyperpolarized (by >10 mV) the $E_M$ of oocytes expressing KCNQ2/3 (Fig. 2i). The data are consistent with quercetin being an important molecular correlate of the KCNQ2/3-activating effects of caper extract.

**Quercetin is an atypical activator of KCNQ2/3 channels**. Quercetin (100 μM) also increased activity of homomeric KCNQ2 and KCNQ3* (KCNQ3-A315T, a mutation that increases KCNQ3 current, facilitating study of the homomeric channel[25]). However, this effect was less voltage-dependent than observed for heteromeric KCNQ2/3, producing less of a shift in activation voltage dependence ($\Delta V_{0.5\text{activation}}$), i.e., $-3.4 \pm 1.2$ mV for KCNQ2 ($n = 4$) and $-4.2 \pm 2.3$ mV for KCNQ3* ($n = 7$) (Fig. 3a, b) and $E_M$ (Fig. 3c). This suggested that quercetin effects on KCNQ2 and KCNQ3 are augmented by KCNQ2-KCNQ3 co-assembly. We did not previously observe this for other individual small molecules, including those from animal, plant or synthetic sources; only combinations of molecules synergized, in KCNQ2 or KCNQ2/3 channels[7,26].

Activation of KCNQ2/3 channels by the anticonvulsant retigabine, which raises the threshold for neuronal excitability, hinges upon negative electrostatic surface potential close to its carbonyl group, a feature present albeit weakly in quercetin (Fig. 2a). This chemical property permits retigabine to activate at a site that requires a specific Trp residue in S5 (W265 in KCNQ3 and its equivalent in other isoforms) (Fig. 3d, e). KCNQ1 lacks this Trp and is retigabine-insensitive[20,27–31]. We previously found that GABA, the metabolite and ketone body β-hydroxybutyrate (BHB), various plant compounds and synthetic glycine derivatives both possess the chemical property noted in retigabine and require the S5 Trp and/or an Arg at the foot of S4 (R242 in KCNQ3 and its equivalents) (Fig. 3d, e) to activate KCNQs. KCNQ1 also possesses the S4 Arg (R243) and this enables KCNQ1 to be activated by some small molecules, including mallotoxin, E-2-dodecenal, and some glycine derivatives (but not GABA or BHB)[7–10,26,32–35].

Here, in contrast to our findings for the small molecules discussed above, we found that quercetin effects on KCNQ2/3 were resilient to mutation of either the S5 W (to L) or the S4 R (to A). Thus, KCNQ2-W236L/KCNQ3-W265L channels and KCNQ2-R213A/KCNQ3-R242A channels responded to quercetin similarly to wild-type KCNQ2/3 channels with respect to negative-shifted voltage dependence (Fig. 3f, g) and $E_M$ (Fig. 3h) (compare to Fig. 2e, f,i).

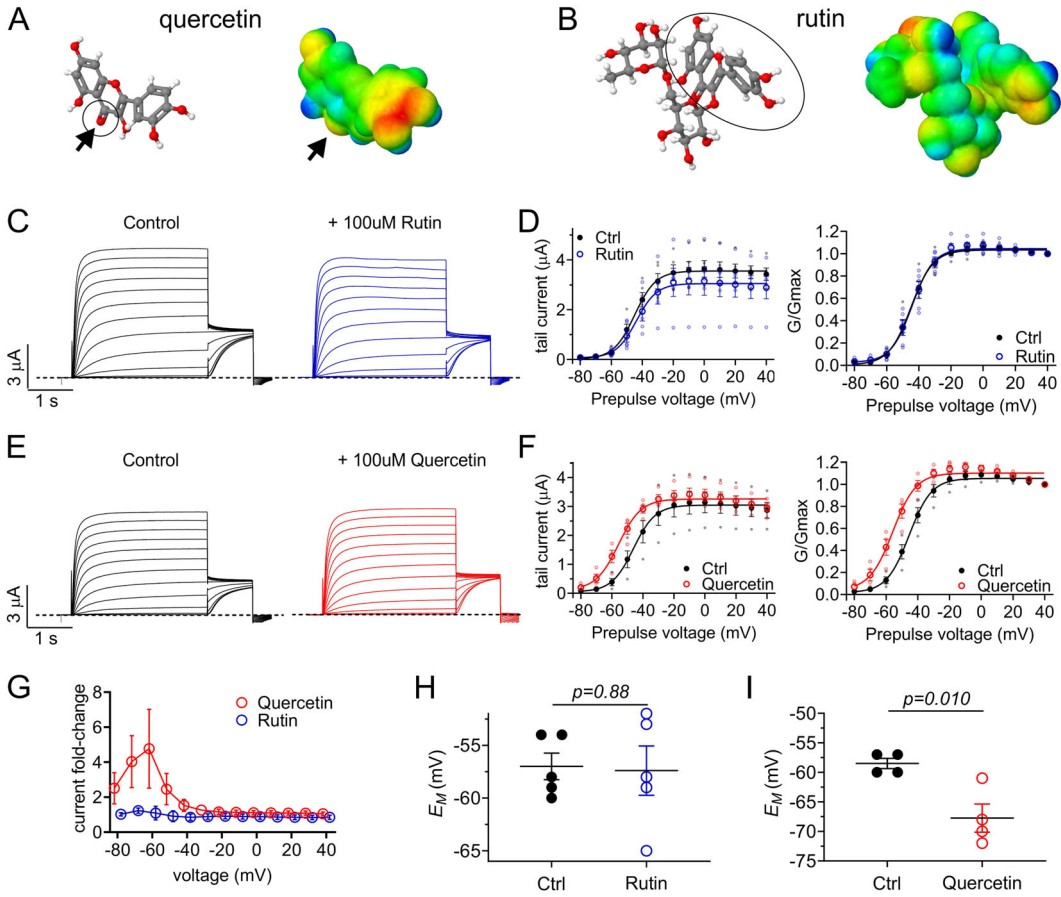

**Fig. 2 Quercetin but not rutin activates KCNQ2/3 channels.** All error bars indicate SEM. $n$ number of oocytes. **a** Chemical structures (red indicates oxygen) and electrostatic surface plot (red, negative; blue, positive) of quercetin. Arrows/circle, carbonyl group. **b** Chemical structures (red indicates oxygen) and electrostatic surface plot of rutin. Circled region indicates quercetin moiety. **c** Mean KCNQ2/KCNQ3 traces in the absence (Control) and presence of 100 μM rutin ($n = 5$). **d** Mean tail current and normalized tail currents (G/Gmax) versus prepulse voltage relationships for the traces as in **c**. **e** Mean KCNQ2/KCNQ3 traces in the absence (Control) and presence of 100 μM quercetin ($n = 4$). **f** Mean tail current and normalized tail currents (G/Gmax) versus prepulse voltage relationships for the traces as in **e**. **g** Current fold-change in KCNQ2/3 tail currents in the presence and absence of quercetin or rutin ($n = 4$–5). **h** Scatter plot of effect of rutin (100 μM) on unclamped membrane potential ($E_M$) of KCNQ2/3-expressing cells as in **c**. Statistical analysis by two-way ANOVA. **i** Scatter plot of effect of quercetin (100 μM) on unclamped membrane potential ($E_M$) of KCNQ2/3-expressing cells as in **e**. Statistical analysis by two-way ANOVA.

We previously found that homomeric KCNQ5 channels are sensitive to GABA, BHB, gabapentin and a range of plant-derived compounds, many at sub-micromolar levels; in contrast, homomeric KCNQ4 was insensitive to GABA, BHB and gabapentin[8,9,32–34]. Here, we found that KCNQ4 current was voltage-independently augmented twofold by 100 μM quercetin (Fig. 3i–k), while KCNQ5 was not activated by 100 μM quercetin, exhibiting instead minor inhibition at voltages positive to −50 mV (Fig. 3l–n). The data in Fig. 3 suggest that quercetin activates KCNQ channels by an atypical mechanism compared to that of other recently studied plant and animal metabolites.

**Quercetin augments KCNQ1 activation and inactivation by an atypical mechanism.** We next focused on KCNQ1. Quercetin exerted striking effects on homomeric KCNQ1, the most obvious being augmentation of the voltage-dependent inactivation that KCNQ1 undergoes (Fig. 4a), a process which is typically only visible by the presence of a "hook" at the start of the tail current. In contrast, quercetin augmented KCNQ1 inactivation such that it was readily visible as current decay, especially at positive voltages, during the prepulse (Fig. 4a). In addition, quercetin negative-shifted the voltage dependence of KCNQ1 activation, by −20 ± 4 mV. This increased KCNQ1 current at voltages more

negative than −30 mV but reduced current at more positive voltages because of the augmented inactivation (Fig. 4b). As the augmented inactivation was strongly voltage-dependent and occurred preferentially at more positive voltages, quercetin was still able to KCNQ1-dependently hyperpolarize $E_M$ (by > 10 mV) (Fig. 4c). In contrast, rutin (100 μM) had no obvious effects on KCNQ1 (Fig. 4a–c).

KCNQ1-R243 lies at the foot of S4 next to the S4-5 linker and we previously found it to be required for activation by contrasting plant compounds including mallotoxin and E-2-dodecenal[7,8,35]. Unlike those compounds, here quercetin was not predicted to bind to KCNQ1-R243, in either the closed or open-state models, using unbiased docking with SwissDock (Fig. 4d, e). Consistent with this, quercetin (100 μM) exerted similar effects on KCNQ1-R243A channels to those on wild-type, i.e., increased inactivation and negative-shifted activation voltage dependence (Fig. 4f, g). Quercetin did not KCNQ1-R243A-dependently shift $E_M$, however, because the baseline positive-shifted voltage dependence of the mutant channel activation precluded influence over $E_M$ even in the presence of quercetin (Fig. 4g, h). The findings were again overall consistent with an atypical modulation mechanism compared to other KCNQ1 modulators.

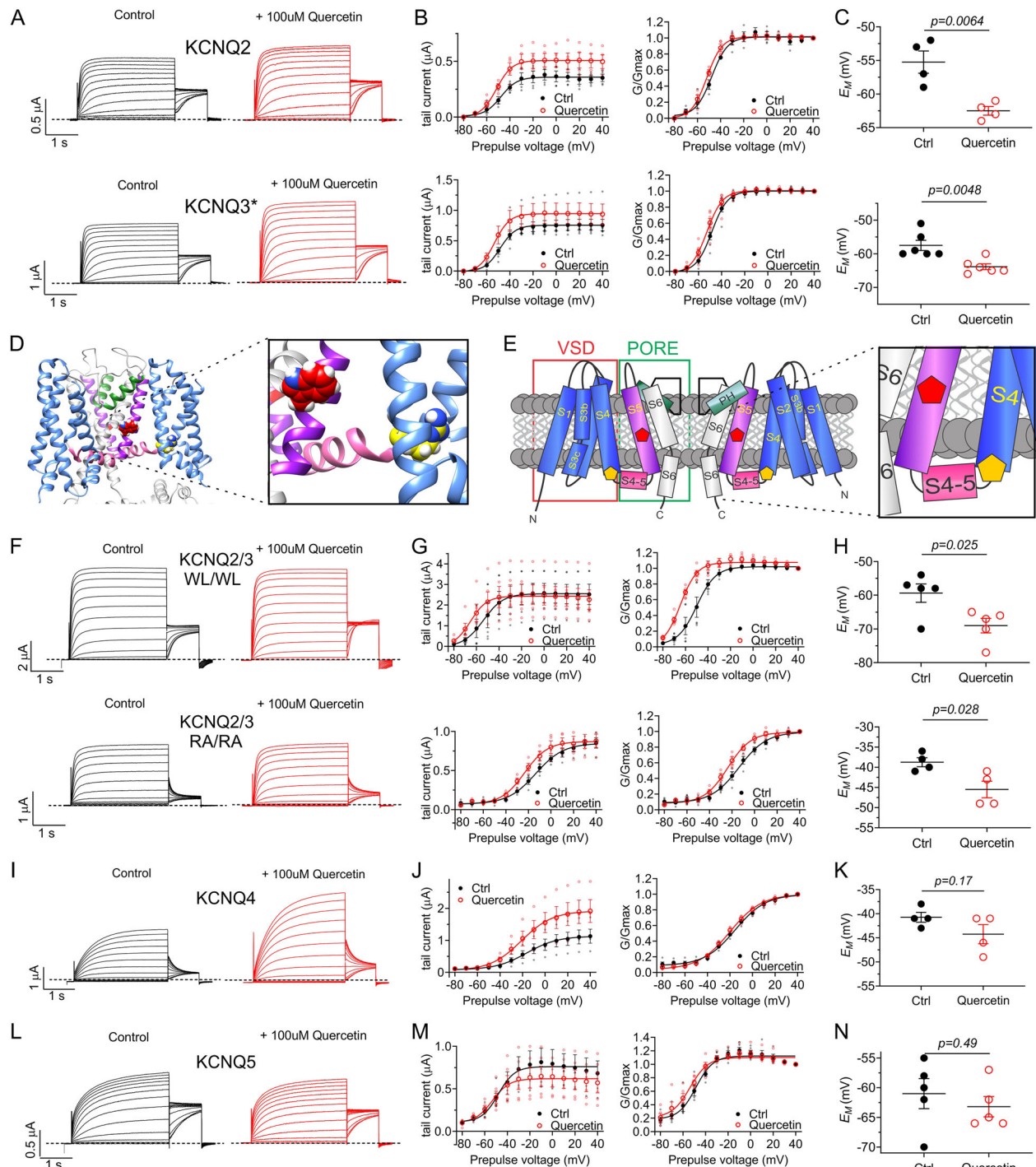

**Fig. 3 Quercetin activates KCNQ2/3 by an atypical mechanism.** All error bars indicate SEM. *n* number of oocytes. **a** Mean TEVC traces of homomeric KCNQ2 and KCNQ3* in the absence (Control) and presence of 100 μM quercetin (*n* = 4–6). **b** Mean tail current and normalized tail currents (G/Gmax) versus prepulse voltage relationships for the traces as in **a**. **c** Scatter plot of unclamped membrane potential ($E_M$) for cells in A. Statistical analysis by two-way ANOVA. **d** Structural model with close-up (boxed) of KCNQ3 based on Cryo-EM structure of *Xenopus* KCNQ1 with KCNQ3-W265 (red) and KCNQ3-R242 (yellow) highlighted. **e** Topological representation with close-up (boxed) of KCNQ3 showing two of the four subunits that comprise a channel, and approximate locations of KCNQ3-W265 (red) and KCNQ3-R242 (yellow). VSD, voltage sensing domain. **f** Mean TEVC traces of heteromeric KCNQ2-W236L/KCNQ3-W265L (KCNQ2/Q3-WL/WL) and heteromeric KCNQ2-R213A/KCNQ3-R242A (KCNQ2/Q3-RA/RA) in the absence (Control) and presence of 100 μM quercetin (*n* = 4–6). **g** Mean tail current and normalized tail currents (G/Gmax) versus prepulse voltage relationships for the traces in **f**. **h** Scatter plot of unclamped membrane potential ($E_M$) for cells in **f**. Statistical analysis by two-way ANOVA. **i** Mean TEVC traces of homomeric KCNQ4 in the absence (Control) and presence of 100 μM quercetin (*n* = 4). **j** Mean tail current and normalized tail currents (G/Gmax) versus prepulse voltage relationships for the KCNQ4 traces as in **i**. **k** Scatter plot of unclamped membrane potential ($E_M$) for KCNQ4-expressing cells as in **i**. Statistical analyses by two-way ANOVA. **l** Mean TEVC traces of homomeric KCNQ5 in the absence (Control) and presence of 100 μM quercetin (*n* = 5). **m** Mean tail current and normalized tail currents (G/Gmax) versus prepulse voltage relationships for the KCNQ5 traces as in **l**. **n** Scatter plot of unclamped membrane potential ($E_M$) for KCNQ5-expressing cells as in **l**. Statistical analyses by two-way ANOVA.

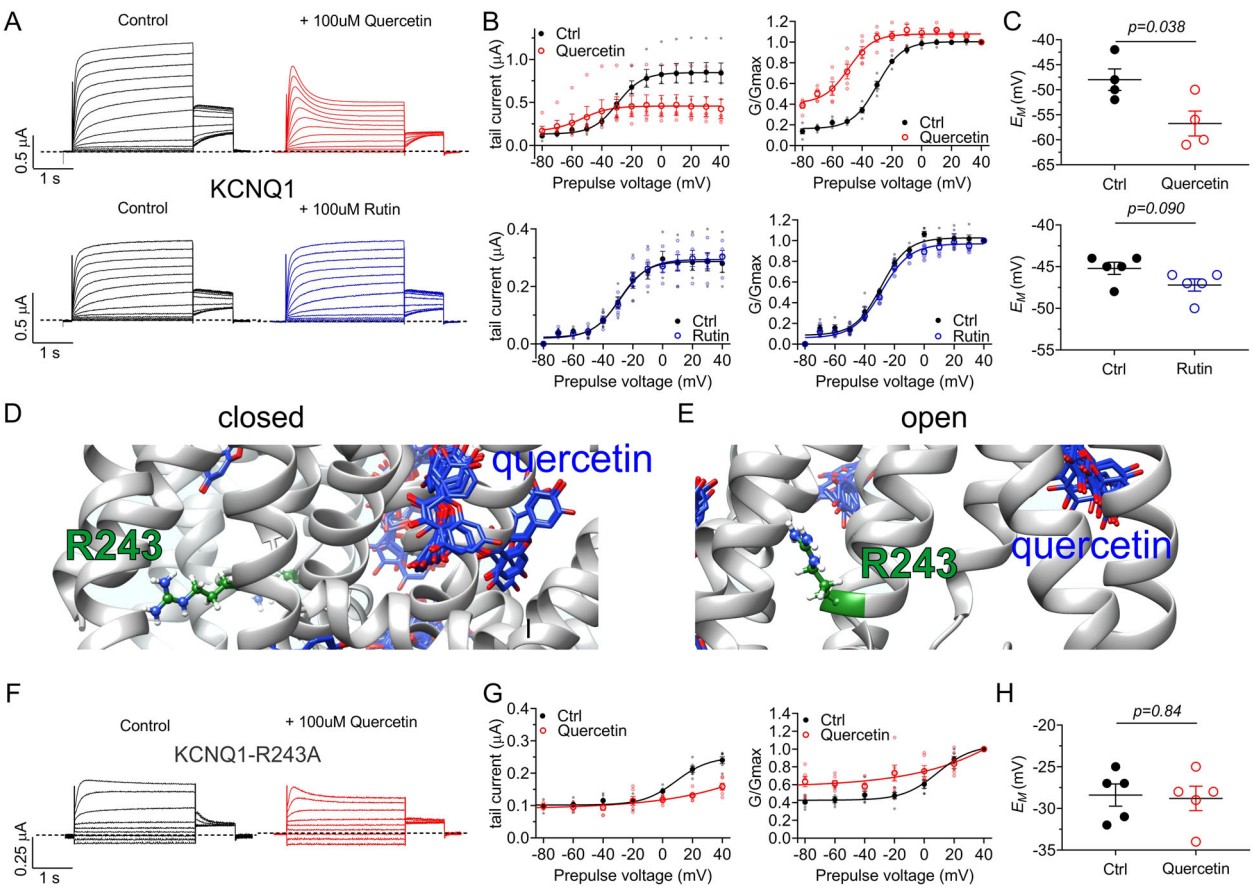

**Fig. 4 Quercetin but not rutin promotes KCNQ1 activation and inactivation.** All error bars indicate SEM. *n* number of oocytes. **a** Mean TEVC traces of homomeric KCNQ1 in the absence (Control) and presence of 100 μM quercetin (*n* = 4) or 100 μM rutin (*n* = 5). **b** Mean tail current and normalized tail currents (G/Gmax) versus prepulse voltage relationships for the traces in **a**. **c** Scatter plot of unclamped membrane potential ($E_M$) for cells in A. Statistical analysis by two-way ANOVA. **d** Human KCNQ1 closed-state structural model showing lack of predicted quercetin binding proximal to R243. **e** Human KCNQ1 open-state structural model showing lack of predicted quercetin binding proximal to R243. **f** Mean TEVC traces of homomeric KCNQ1-R243A in the absence (Control) and presence of 100 μM quercetin (*n* = 5). **g** Mean tail current and normalized tail currents (G/Gmax) versus prepulse voltage relationships for the traces in **f**. **h** Scatter plot of unclamped membrane potential ($E_M$) for cells in **f**. Statistical analyses by two-way ANOVA.

To quantify the effects of quercetin on KCNQ1 activation rate we employed a protocol that enabled quantification of peak tail current at −120 mV after recovery from inactivation and before closure, in response to depolarizing prepulses of varying durations (Fig. 5a, b). Fitting of these data with a single exponential function indicated that quercetin speeds KCNQ1 activation 5.7-fold (Fig. 5c) (from $\tau = 153 \pm 4$ to $27 \pm 2$ ms) (*n* = 5). Fitting deactivation traces at −120 mV (Fig. 5a) with a single exponential function, quercetin was found to slow KCNQ1 deactivation 2.5-fold, (from $\tau = 70 \pm 4$ to $181 \pm 25$ ms) (*n* = 5). These data are suggestive of the possibility that quercetin stabilizes the open state of KCNQ1 and/or destabilizes its closed state. It is, however, possible that the altered inactivation and/or rate of recovery from inactivation of quercetin influence the effects we observe on activation or deactivation.

Employing a simple voltage protocol incorporating a brief (20 ms) inactivation recovery step to −120 mV between two +40 mV pulses, the inactivation recovery "hook" was readily apparent after the recovery step (solid arrows, Fig. 5d). When returning to −80 mV after the second +40 mV pulse, larger tail currents were observed in the presence of quercetin, probably due to a combination of a greater proportion of inactivated channels and slower deactivation in the presence of quercetin (open arrows, Fig. 5d). Reverting to the standard voltage protocol, quantification of the normalized inactivation recovery hook versus prepulse

voltage revealed a >2-fold increase in the relative size of the hook in the presence of quercetin, also indicating increased inactivation (Fig. 5e, f).

**Quercetin is predicted to have multiple binding sites on KCNQ1.** In silico docking studies suggested multiple potential quercetin binding locations in KCNQ1. One striking prediction was that quercetin docked to the extracellular surface of the VSD in the closed state (but not the open state) model of human KCNQ1[36] (Fig. 6a, b). The closest S4 charged residues to the predicted quercetin docking site were R228 and R231 (Fig. 6c). We next conducted studies of docking to the cryo-electron microscopy (cryo-EM)-derived human KCNQ1-calmodulin structure[37]. Strikingly, of 35 predicted quercetin docking clusters, the two with the most negative $\Delta G$ values (−8.56 and −8.04 kJ/mol) showed predicted binding close to and/or H-bonding with the R231 sidechain (Fig. 6d, e). We therefore tested the effects of neutralizing R228 and R231 by mutation to alanine. The R228A mutation positive-shifted the voltage dependence of KCNQ1 activation as we previously reported[38]. More strikingly, the R228A substitution removed the ability of quercetin to augment KCNQ1 activation, instead resulting in mild inhibition at positive membrane potentials (Fig. 6f, g). Quantification of the normalized inactivation recovery hook in the tail current (Fig. 6d) revealed a ~twofold increase with quercetin (Fig. 6h), while there

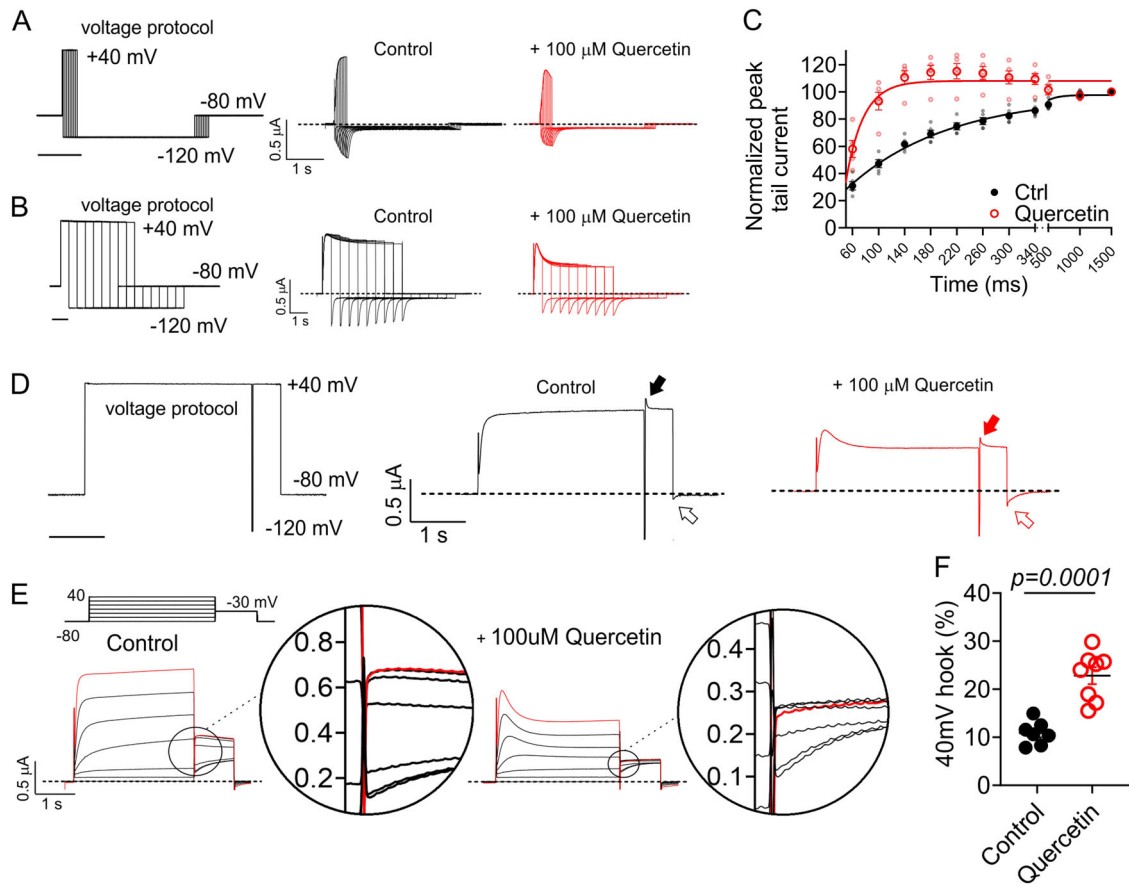

**Fig. 5 Quercetin speeds KCNQ1 activation and inactivation.** All error bars indicate SEM. $n$ number of oocytes. **a** Voltage protocol and mean TEVC current traces of KCNQ1 comparing the relationship between shorter prepulse durations and normalized tail current at −80 mV ($n = 4$) in the absence and presence of 100 µM quercetin. **b** Voltage protocol and mean TEVC current traces of KCNQ1 comparing the relationship between longer prepulse durations and normalized tail current at −80 mV ($n = 4$) in the absence and presence of 100 µM quercetin. **c** Mean normalized peak tail current versus prepulse duration in the absence (Ctrl) or presence of 100 µM quercetin quantified from traces in **a**, **b** and fit with a single exponential function ($n = 4$). **d** Voltage protocol and mean TEVC current traces of KCNQ1 comparing the tail current recovery "hook" currents at +40 mV (solid arrow) and −80 mV (open arrow) in the absence and presence of 100 µM quercetin ($n = 5$). **e** Close-up view of inactivation recovery "hooks" taken from initial tail current portion of mean KCNQ1 current traces in the absence (Control) and presence of 100 µM quercetin. Red line indicates −30 mV tail pulse after +40 mV prepulse, used for quantification in **f**. **f** Mean fractional KCNQ1 hook current in the absence (Control) or presence of 100 µM quercetin at −30 mV following a +40 mV prepulse, quantified from traces as in **e** ($n = 7$).

was no change in $E_M$ (Fig. 6i). The data are consistent with quercetin still binding to KCNQ1-R228 and augmenting its inactivation but being unable to augment activation. KCNQ1-R231A channels are constitutively active[38] without visible evidence of inactivation (no hook apparent in tail current and no visible prepulse inactivation). It is possible that KCNQ1-R231A channels are also constitutively inactivated, although we found previously that the R231A mutation removes the voltage-dependent inactivation of F340W-$I_{Ks}$ channels, which are normally constitutively activated but exhibit voltage-dependent inactivation[39]. Here, quercetin exerted effects neither on KCNQ1-R231A current magnitude, $V_{0.5activation}$ (Fig. 6j, k), nor $E_M$ of KCNQ1-R231A-expressing oocytes (Fig. 6l). The data suggest that R228 is required for quercetin activation of, but not binding to or inactivation of, KCNQ1; R231 appears influential in quercetin binding and/or its functional effects, the functional data being consistent with the predicted docking site near to R228 and R231 and possible H-bonding to the R231 sidechain.

We next focused on another predicted quercetin binding site, F340 on S6 (Fig. 7a). F340 is particularly interesting because we previously identified this residue as a hub for KCNQ1 gating processes, finding it to be highly influential in both activation and

inactivation[39]. Here, we found that KCNQ1-F340V channels are insensitive to quercetin (100 µM), with no effects on current magnitude, voltage dependence of activation, inactivation (Fig. 7b–d) or $E_M$ (Fig. 7e). Thus, consistent with the docking prediction, KCNQ1-F340 is important for the functional effects and/or physical interaction of quercetin.

**Quercetin augments KCNQ1-KCNE1 activity without inducing inactivation.** KCNQ1 forms heteromeric channel complexes in cardiac myocytes with the KCNE1 ancillary subunit that are essential for human ventricular repolarization, and are also important for auditory function because of their role in regulating endolymph secretion in the inner ear[40,41]. KCNE1 slows and positive-shifts the voltage dependence of KCNQ1 activation, yet also increases microscopic and macroscopic current magnitude and eliminates inactivation[21,22,42–44].

Quercetin (100 µM) augmented KCNQ1-KCNE1 current across a wide voltage range and most prominently at −30 mV (Fig. 8a, b) and hyperpolarized $E_M$ by −12 mV (Fig. 8c). Unguided docking studies using KCNQ1-KCNE1 model structures[36] again predicted quercetin binding near the top of S4, and in some poses quercetin H-bonded with R228 (in the open-state

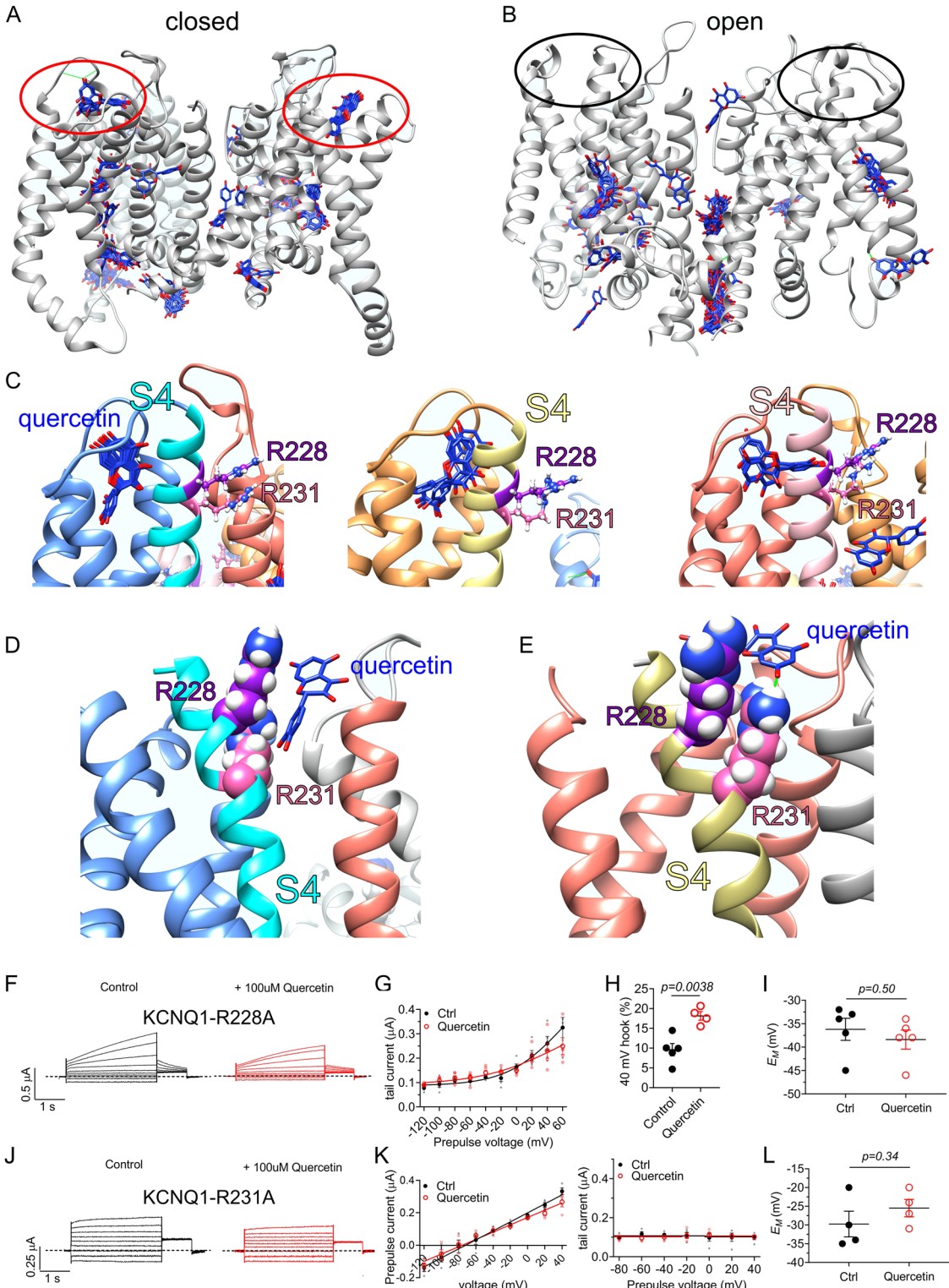

model) and R231 (in the closed-state model) (Fig. 8d–f). Accordingly, quercetin had little-to-no effects on KCNQ1-R228A channels, except minor inhibition at the most depolarized potentials (with no evidence of inactivation in wild-type or R228A KCNQ1-KCNE1) (Fig. 8g–i), and no effects on KCNQ1-R231A (Fig. 8j–l).

Unlike in homomeric KCNQ1, quercetin was not predicted to dock to F340 in KCNQ1-KCNE1, possibly because KCNE1 alters the environment around this residue or impedes the normal

quercetin binding position. We therefore used mutation of F340 to probe a different property of quercetin, i.e., its effects on inactivation. We previously found that in contrast to wild-type KCNQ1-KCNE1, channels formed by KCNE1 and KCNQ1-F340W exhibit both constitutive activation and robust, voltage-dependent inactivation that is readily visible as current decay during the prepulse at more depolarized voltages. Homomeric KCNQ1-F340W channels behave qualitatively similarly to this, but with slower, less prominent inactivation[39,45]. Here, quercetin

**Fig. 6 Quercetin requires extracellular S4 charged residues R228 and R231 for KCNQ1 current augmentation.** All error bars indicate SEM. $n$ number of oocytes. **a** Human KCNQ1 closed-state structural model[36] showing predicted quercetin binding to the top of the VSD (red circles). **b** Human KCNQ1 open-state structural model[36] showing lack of predicted quercetin binding to the top of the VSD (black circles). **c** Human KCNQ1 closed-state structural model close-ups[36] showing predicted quercetin binding to the top of 3 of the VSDs. **d** Close-up of top of one VSD from the cryo-EM-derived human KCNQ1-calmodulin structure showing predicted docking of quercetin close to the R231 sidechain. **e** Close-up of top of another VSD from the cryo-EM-derived human KCNQ1-calmodulin structure showing predicted H-binding (green line) of quercetin to the R231 sidechain. **f** Mean TEVC current traces showing effects of quercetin (100 μM) on KCNQ1-R228A ($n = 5$). **g** Mean peak tail current versus prepulse voltage relationship for the traces as in **f** ($n = 5$). **h** Scatter plot of mean normalized tail hook for the traces as in **f** ($n = 4$–5). **i** Scatter plot of unclamped membrane potential ($E_M$) for cells as in **f** ($n = 5$). Statistical analysis by two-way ANOVA. **j** Mean TEVC current traces showing effects of quercetin (100 μM) on KCNQ1-R231A ($n = 5$). **k** Mean peak prepulse current and tail current versus prepulse voltage relationships for the traces as in **j** ($n = 5$). **l** Scatter plot of unclamped membrane potential ($E_M$) for cells as in **j** ($n = 4$). Statistical analysis by two-way ANOVA.

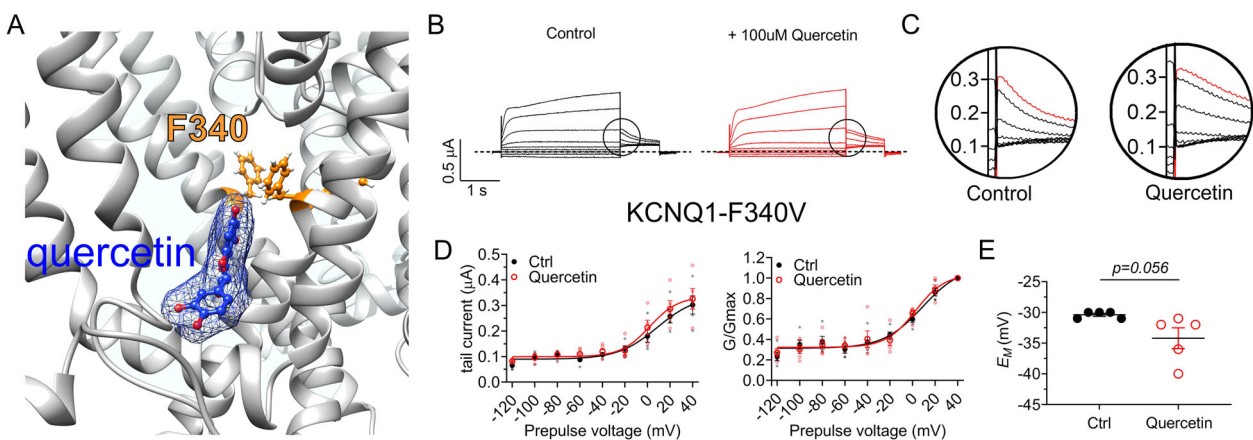

**Fig. 7 Quercetin requires S6 residue F340 for modulation of KCNQ1 activation and inactivation.** All error bars indicate SEM. $n$ number of oocytes. **a** Human KCNQ1 open-state structural model close-up[36] showing predicted quercetin binding to F340. **b** Mean TEVC current traces showing effects of quercetin (100 μM) on KCNQ1-F340V ($n = 5$). **c** Close-up of inactivation recovery "hook" currents from traces as in **b**. **d** Mean peak tail current (left) and normalized peak tail current (G/Gmax) (right) versus prepulse voltage relationships for the traces as in **b** ($n = 5$). **e** Scatter plot of unclamped membrane potential ($E_M$) for cells as in **b** ($n = 5$). Statistical analysis by two-way ANOVA.

(100 μM) speeded activation of homomeric KCNQ1-F340W current, especially at more depolarized voltages, causing peak current to be reached much earlier (Fig. 8m) yet also suppressed currents, again more so at more depolarized voltages suggesting enhancement of inactivation. Indeed, slow current decay was evident during the prepulse only after quercetin application (Fig. 8m, n). $E_M$ was unaffected (Fig. 8o) as the majority of constitutive current at rest was preserved after quercetin application (Fig. 8n). In contrast, quercetin uniformly augmented KCNQ1-F340W/KCNE1 current across the voltage range (Fig. 8p–r). This could hypothetically arise because inactivation is at its fullest permissible extent at baseline in KCNQ1-F340W-KCNE1 channels; in this scenario quercetin would be modulating solely activation. The alternative, and we consider more likely, explanation, which we previously suggested based on their disparate voltage dependence, is that inactivation is uncoupled from activation in KCNQ1-F340W-KCNE1 channels[39]. This could mean that quercetin only modulates inactivation indirectly via coupling to activation in, e.g., wild-type KCNQ1.

**Quercetin augments voltage-dependent activation in KCNQ1-KCNE3 channels.** KCNE3 converts KCNQ1 to a constitutively active, non-inactivating channel that is important in intestinal and airway epithelia[46–48]. Quercetin augmented KCNQ1-KCNE3 current between −40 and +40 mV by introducing a voltage-dependent component across this voltage range that superimposed upon the constitutively active component (Fig. 9a, b). Because the latter component was unchanged, the $E_M$ was relatively unchanged by quercetin (100 μM) (Fig. 9c). As we observed for KCNQ1 and KCNQ1-KCNE1 channels, in the human

KCNQ1-KCNE3 model structure[49] docking simulations quercetin was predicted to bind near the top of the VSD, close to R228 and R231 (Fig. 9d, e). Quercetin also docked to the top of the VSD in the cryo-EM-derived structure of human KCNQ1-PIP$_2$-calmodulin/KCNE3[37], notably being predicted to H-bond to the R231 sidechain. Out of 27 KCNQ1 predicted binding clusters in the KCNQ1-PIP$_2$-calmodulin/KCNE3 structure, the top three in terms of most negative ΔG values (−8.75, −8.26 and −7.82 kJ/mol) positioned quercetin close to and/or H-binding with the R231 sidechain (Fig. 9f). Quercetin modulated KCNQ1-R228A/KCNE3 channels in a similar manner to effects on wild-type KCNQ1-KCNE3, demonstrating that R228 is relatively unimportant for quercetin effects on KCNQ1-KCNE3 (Fig. 9g–i). In contrast, quercetin had no effect on KCNQ1-R231A/KCNE3 channel activity, demonstrating the importance of this residue for quercetin binding and/or functional effects (Fig. 9j–l), consistent with the docking results.

**Combinatorial effects of quercetin and other plant compounds.** Though the most striking changes in activity were seen in KCNQ2/3 and KCNQ1 with and without KCNE1 or KCNE3, quercetin also had effects on the other homomeric KCNQ channels (Fig. 3), shifting their $V_{0.5activation}$ and augmenting activity (Fig. 10a, b). Further, despite the inability of rutin to activate heteromeric KCNQ2/3 and homomeric KCNQ1, it modestly augmented KCNQ1/KCNE1 and KCNQ3* activity and induced a −4.5 ± 0.9 mV shift in KCNQ3* $V_{0.5activation}$; rutin also hyperpolarized the $E_M$ of cells expressing either KCNQ1/KCNE1 or KCNQ3* (Fig. 10c).

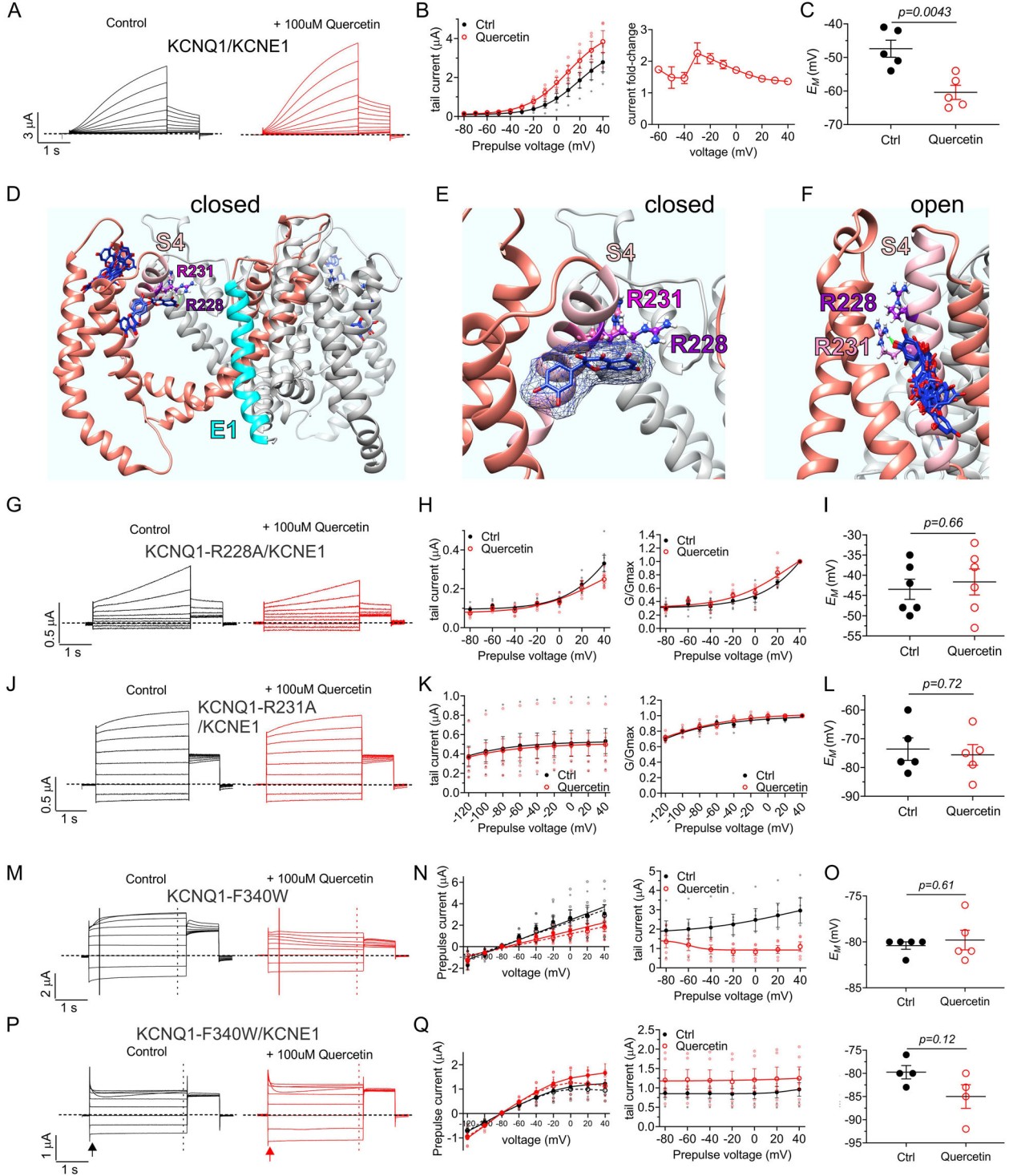

To probe potential synergistic effects of rutin and quercetin, we tested a combination of 10 μM quercetin and 10 μM rutin on KCNQ1/E1 and KCNQ3*. There was no evidence of synergy or even additive effects between rutin and quercetin on KCNQ1/KCNE1 (Fig. 10d, e). In contrast, rutin and quercetin augmenting effects on KCNQ3* activity were additive (Fig. 10d, f).

Previously, we showed that E-2-dodecenal, a fatty aldehyde found in cilantro, activates KCNQ1 via R243[8], a binding site that quercetin is not predicted to bind to (Fig. 10g), suggesting the possibility of synergy between the two compounds. We did not

observe synergistic augmentation of KCNQ1 current by the combination of either 10 + 10 μM, or 100 + 100 μM (quercetin + E-2-dodecenal), either by overall current potentiation (Fig. 10h–j) or $\Delta V_{0.5\text{activation}}$ (Fig. 10h, k). In contrast, and despite E-2-dodecenal not enhancing KCNQ1 inactivation per se[8], E-2-dodecenal and quercetin synergistically enhanced KCNQ1 inactivation (Fig. 10h, l). Specifically, with 10 μM of each compound, we observed augmented inactivation equivalent to that of 100 μM quercetin alone (24% and 23%, respectively); the combination of 100 μM E-2-dodecenal and quercetin increased hook size to 58% of the total tail current (Fig. 10l).

**Fig. 8 Quercetin requires S4 charged residues R228 and R231 for augmentation of KCNQ1/KCNE1 current.** All error bars indicate SEM. *n* number of oocytes. **a** Mean current traces showing effects of quercetin (100 μM) on KCNQ1/KCNE1 (*n* = 5). **b** Mean peak tail current and current fold-change versus prepulse voltage for traces as in **a** (*n* = 5). **c** Scatter plot of unclamped membrane potential ($E_M$) for cells as in A (*n* = 5). Statistical analyses by two-way ANOVA. **d** Human KCNQ1/KCNE1 closed-state structural model[36] showing predicted quercetin binding to the top of the VSD. **e** Human KCNQ1 closed-state structural model close-up[36] showing predicted quercetin binding to the top of the VSD. **f** Human KCNQ1 open-state structural model close-up[36] showing predicted quercetin binding to the VSD. **g** Mean current traces showing effects of quercetin (100 μM) on KCNQ1-R228A/KCNE1 (*n* = 6). **h** Mean peak and normalized peak tail current (G/Gmax) versus prepulse voltage for traces as in **g** (*n* = 6). **i** Scatter plot of unclamped $E_M$ for cells as in **g** (*n* = 6). Statistical analyses by two-way ANOVA. **j** Mean current traces showing effects of quercetin (100 μM) on KCNQ1-R231A/KCNE1 (*n* = 5). **k** Mean peak and normalized peak tail current (G/Gmax) versus prepulse voltage for the traces as in **j** (*n* = 5). **l** Scatter plot of unclamped membrane potential ($E_M$) for cells as in **j** (*n* = 5). Statistical analysis by two-way ANOVA. **m** Mean current traces showing effects of quercetin (100 μM) on KCNQ1-F340W (*n* = 5). **n** Mean peak and normalized peak tail current (G/Gmax) versus prepulse voltage for the traces as in **m** (*n* = 5). **o** Scatter plot of unclamped $E_M$ for cells as in **m** (*n* = 5). Statistical analysis by two-way ANOVA. **p** Mean current traces showing effects of quercetin (100 μM) on KCNQ1-F340W/KCNE1 (*n* = 5). **q** Mean peak and normalized peak tail current (G/Gmax) versus prepulse voltage for the traces as in **p** (*n* = 5). **r** Scatter plot of unclamped $E_M$ for cells as in **p** (*n* = 5). Statistical analysis by two-way ANOVA.

## Discussion

Archaeological evidence for human caper consumption dates back as far as Mesolithic soil deposits in Syria thought to be more than 10,000 years old, and late Stone Age cave dwellings in the Greek Peloponnesian peninsula and Israel[50]. Capers are also in current use or study for their anti-helminthic, anti-cancer, anti-diabetic and anti-inflammatory properties and possible circulatory and gastrointestinal benefits[51].

Here we found that a methanolic extract of pickled capers—the form of capers most commonly available in the United States—KCNQ-dependently hyperpolarizes cells. Pickled capers are used throughout the world for a variety of culinary purposes. In the U. S. they are often added for flavoring to smoked salmon, pasta and other dishes. Pickling promotes conversion of rutin to quercetin, the ingredient that we found to be an efficacious KCNQ channel activator. This contributes to pickled capers being the richest known "natural" source of quercetin, with a maximum reported concentration of 520 mg/100 g for canned capers, compared to a maximum of 323 mg/100 g quercetin for raw capers, although mean reported quercetin values are more similar between the two forms, suggesting variations in the pickling or canning process could affect content[23,51]. Although a 1:100 dilution of pickled caper extract was sufficient to hyperpolarize cells because of the strong influence of KCNQ channels on $E_M$, it is important to note that quercetin was much less potent than other plant-derived KCNQ channel activators, such as E-2-dodecenal from, e.g., cilantro (*Coriandrum sativum*), a plant that also contains quercetin[23]. This may be because quercetin binds to more than one, low-affinity site on each KCNQ channel subunit, as suggested by docking studies herein, whereas E-2-dodecenal appears instead to have one high-affinity site[8]. Nevertheless, the ability of quercetin to augment KCNQ1, KCNQ1-KCNE1 and KCNQ1/KCNE3 activity apparently by binding to the top of the VSD (and especially to R231) and/or F340 in S6 suggests that quercetin represents a novel chemical space for medicinal chemistry development of future KCNQ channel activators that adopt a different augmentation mechanism than other well-known KCNQ activators[7–9,20,26,32,33,35,52,53]; this type of development would also be targeted toward avoiding hERG inhibition, a property previously linked to quercetin although the binding site and mechanism were not determined[54].

Because of the divergent predicted KCNQ channel binding sites of quercetin and E-2-dodecenal (supported by mutagenesis and electrophysiology studies) we investigated their potential synergy and found synergistic effects on KCNQ1 inactivation but not activation. As E-2-dodecenal does not enhance KCNQ1 inactivation on its own[8], this supports a mechanistic model in which quercetin enhances inactivation via the specific mechanism by which it enhances KCNQ1 activation, which differs (at least in binding site) from that induced by E-2-dodecenal. With a combination of the two compounds, it appears that the activation-enhancing effects of E-2-dodecenal can synergize with quercetin to enhance activation-linked inactivation. The concept that quercetin enhances KCNQ1 inactivation only indirectly via enhancing effects on activation is supported by the inability of quercetin to induce inactivation in non-inactivating KCNQ1-R231A, KCNQ1/KCNE1 and KCNQ1/KCNE3 complexes, while being able to enhance inactivation in the inactivating mutant KCNQ1-F340W/KCNE1 channel. This mechanistic nuance is potentially important for the physiological effects of quercetin in vivo: KCNQ1 is thought not to exist as a homomeric channel in vivo, but rather as heteromeric channels with various KCNE subunits that protect it against inactivation, such as KCNE1, 2 and 3[18]. Therefore, the KCNE subunits protect KCNQ1 from inhibition by quercetin yet permit the current-enhancing effects of quercetin.

Although the activation-enhancing effects of quercetin and plant compounds such as rutin and E-2-dodecenal were not additive for KCNQ1, they were for KCNQ3*. Thus, while quercetin itself may be at insufficient quantities in capers to exert strong influence on KCNQ channels in vivo merely from ingestion of typical quantities of capers, it is possible that co-ingestion with other plants containing compounds, the effects of which summate or synergize with those of quercetin (e.g., in capers), could underlie some KCNQ-dependent therapeutic effects. It is also notable that quercetin is the most prevalent flavonoid in the human diet, being found in commonly eaten foods including apples, berries, celery, chili peppers, kale and onions to name a few; it is also present in other commonly consumed food products such as green tea and red wine[23,55]. In addition, quercetin is a highly pervasive molecule, freely traveling though cell membranes and accessing cellular organelles including the nucleus[56]. Given these two factors, it is possible that quercetin accumulates in our bodies over time and beneficially enhances KCNQ currents.

In summary, quercetin activates KCNQ channels by an atypical mechanism that may lead the way to a new chemical space for novel modes of pharmacological KCNQ activation. Its relatively low potency for KCNQ modulation may limit the therapeutic usefulness of quercetin, although higher doses could potentially be tolerated and it was recently concluded that addition of quercetin to food appears to be safe[57]. In addition to serving as a novel chemical direction for KCNQ modulation, quercetin may be a useful tool to study, e.g., KCNQ1 inactivation. Further, given the pervasive and essential roles of KCNQ channels throughout the body, and the many intersections between quercetin and

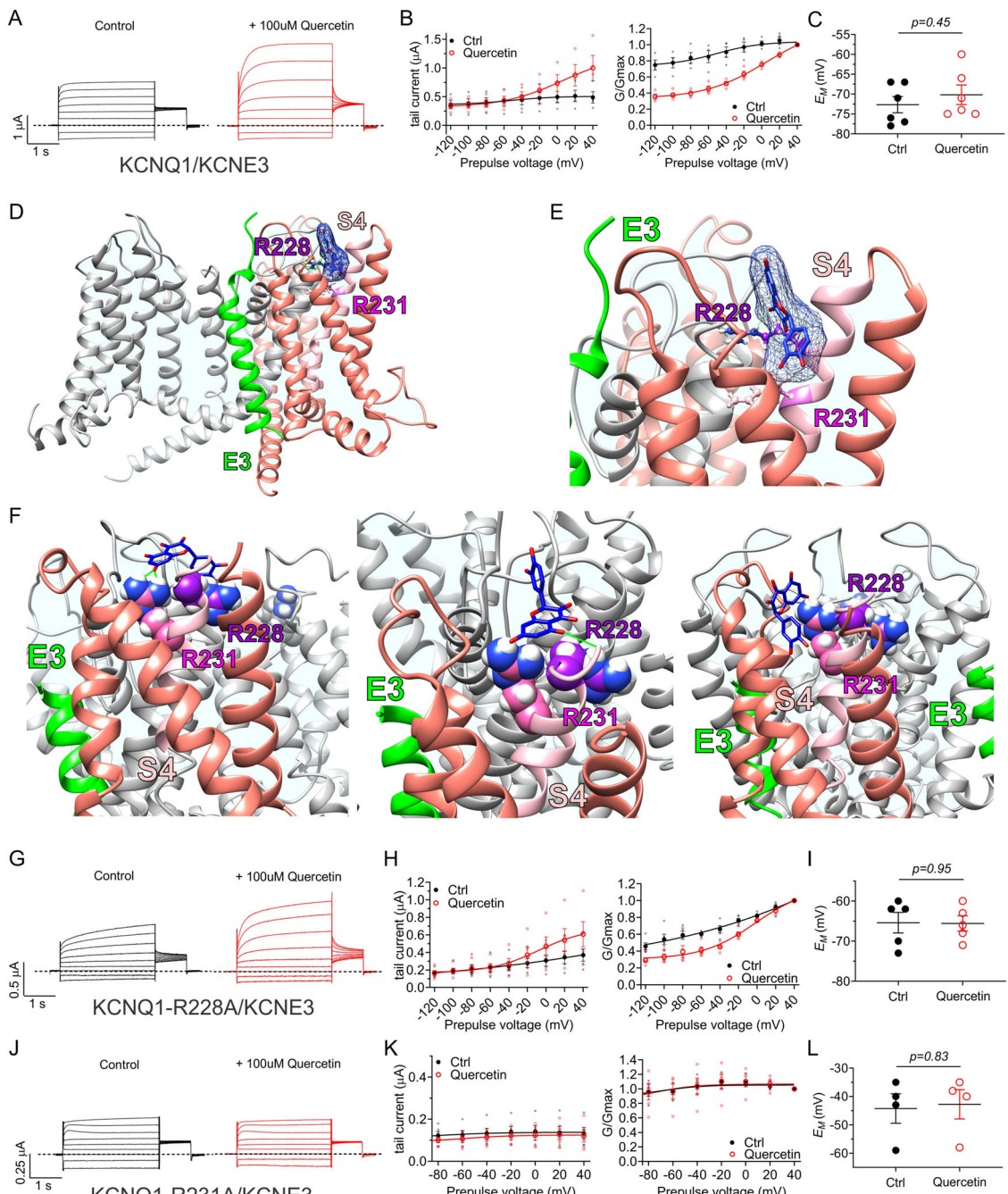

**Fig. 9 Quercetin requires S4 charged residue R231 for augmentation of KCNQ1/KCNE3 current.** All error bars indicate SEM. *n* number of oocytes.
**a** Mean TEVC current traces showing effects of quercetin (100 μM) on KCNQ1/KCNE3 (*n* = 6). **b** Mean peak tail current (left) and normalized peak tail current (G/Gmax) (right) versus prepulse voltage relationships for the traces as in **a** (*n* = 6). **c** Scatter plot of unclamped membrane potential (*E*$_M$) for cells as in **a** (*n* = 6). Statistical analyses by two-way ANOVA. **d** Human KCNQ1/KCNE3 structural model[49] showing predicted quercetin binding to the top of the VSD. **e** Human KCNQ1/KCNE3 structural model[49] close-up showing predicted quercetin binding to the top of the VSD. **f** Three predicted docking clusters at the top of the VSD of the cryo-EM-derived human KCNQ1-PIP$_2$-calmodulin/KCNE3 structure[37] showing docking of quercetin close to and/or hydrogen bonding (green line) with the R231 sidechain. **g** Mean TEVC current traces showing effects of quercetin (100 μM) on KCNQ1-R228A/KCNE3 (*n* = 5). **h** Mean peak tail current (left) and normalized peak tail current (G/Gmax) (right) versus prepulse voltage relationships for the traces as in **g** (*n* = 5). **i** Scatter plot of unclamped membrane potential (*E*$_M$) for cells as in **g** (*n* = 5). Statistical analyses by two-way ANOVA. **j** Mean TEVC current traces showing effects of quercetin (100 μM) on KCNQ1-R231A/KCNE3 (*n* = 4). **k** Mean peak tail current (left) and normalized peak tail current (G/Gmax) (right) versus prepulse voltage relationships for the traces as in **j** (*n* = 4). **l** Scatter plot of unclamped membrane potential (*E*$_M$) for cells as in **j** (*n* = 4). Statistical analyses by two-way ANOVA.

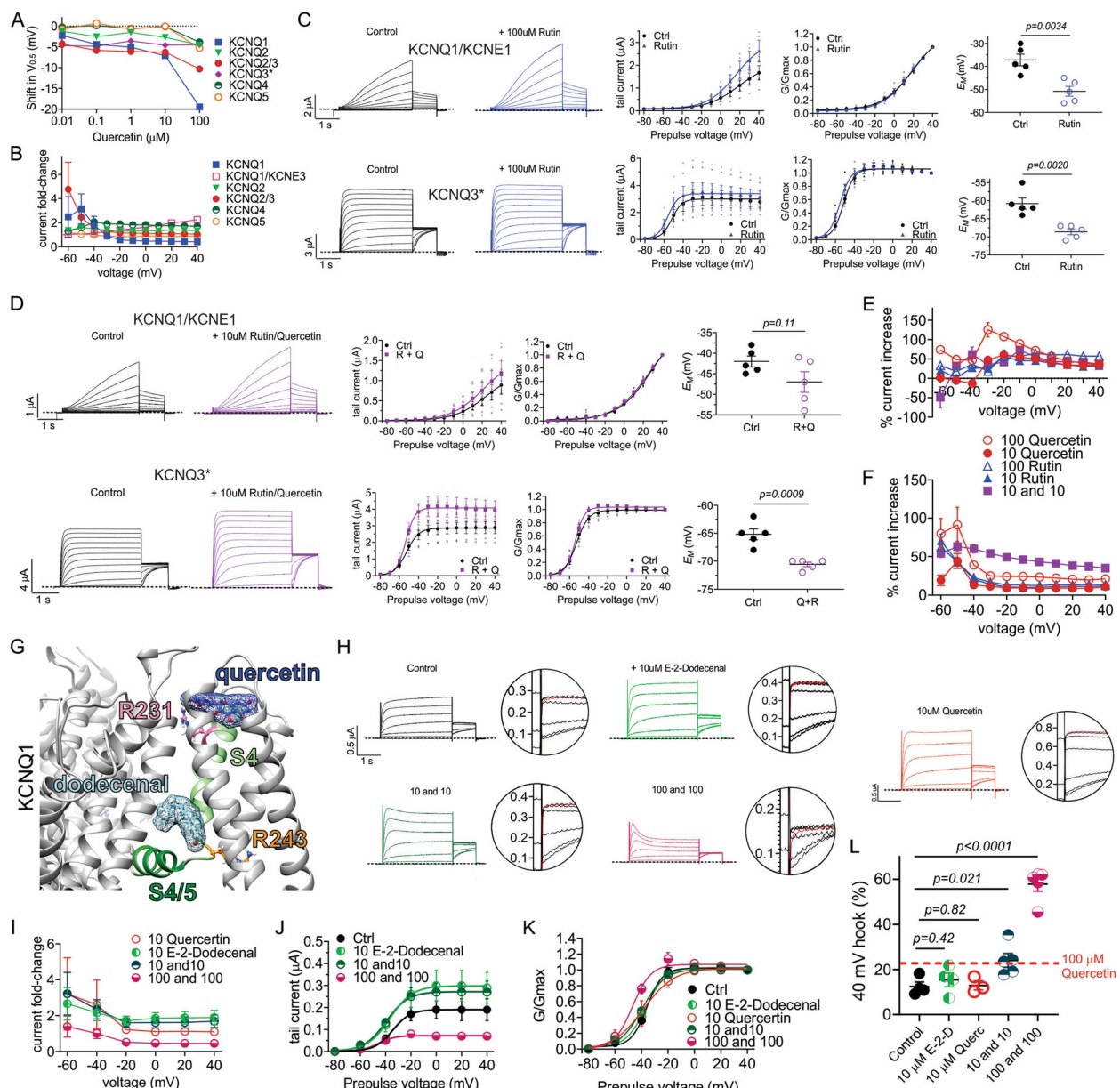

**Fig. 10 Quercetin and cilantro aldehyde E-2-dodecenal combine to synergistically inactivate KCNQ1.** All error bars indicate SEM. $n$ number of oocytes. **a** Quercetin dose response for channels indicated, quantified by shift in midpoint voltage dependence of activation ($\Delta V_{0.5 \text{activation}}$). **b** Mean current fold-changes for channels indicated, quantified by the quotient of the tail currents in the presence and absence of 100 μM quercetin. **c** Mean traces, tail and normalized tail current (G/Gmax) and $E_M$ as indicated for KCNQ1/KCNE1 and KCNQ3* in the absence (Ctrl) and presence of 100 μM rutin ($n = 5$). Statistical analysis by two-way ANOVA. **d** Mean TEVC traces, tail and normalized tail current (G/Gmax) and $E_M$ as indicated for KCNQ1/KCNE1 and KCNQ3* in the absence (Ctrl) and presence (R + Q) of 10 μM rutin plus 10 μM quercetin ($n = 5$). Statistical analysis by two-way ANOVA. **e** Mean KCNQ1/KCNE1 current increase (%) induced by rutin, quercetin or the combination of both (values in micromolar). **f** Mean KCNQ3* current increase (%) induced by rutin, quercetin or the combination of both (values in micromolar). **g** Human KCNQ1 structural model showing predicted quercetin and E-2-dodecenal binding locations. **h** Mean TEVC traces for KCNQ1 in the absence (Control) or presence of 10 μM E-2-dodecenal, 10 μM quercetin, a combination of 10 μM E-2-dodecenal and 10 μM quercetin (10 and 10), or a combination of 100 μM E-2-dodecenal and 100 μM quercetin (100 and 100). Close-up of the tail current inactivation "hooks" are to the right of each trace ($n = 3–5$). **i** Mean current fold-changes for KCNQ1, quantified by the quotient of the tail currents in the presence and absence of the conditions tested in H ($n = 3–5$). **j** Mean tail current versus prepulse voltage for traces as in **h** ($n = 3–5$). **k** Normalized tail currents (G/Gmax) versus prepulse voltage for traces as in **h** ($n = 3–5$). **l** Mean fractional KCNQ1 hook current versus total tail current in the absence (Control) or presence of conditions tested in **h** at −30 mV following a +40 mV prepulse, quantified from traces as in **h** ($n = 3–5$).

KCNQ channels in terms of tissue sites of action/expression, KCNQ modulation may contribute to the medicinal effects of quercetin and plant components such as capers, which continue to be ingested globally on a daily basis and have been used as folk medicine for thousands of years.

## Materials and methods

**Preparation of caper extract.** Pickled capers (*Capparis spinosa*) were sourced from Trader Joe's (Irvine, CA, US), rinsed in deionized water to remove excess sodium chloride and homogenized with a bead mill using porcelain beads in 50 ml tubes (Omni International, Kennesaw, GA, US). We performed a methanolic extraction (80% methanol/20% water) on the caper homogenate for 48 h at room

temperature on a rocking platform, with occasional inversion of the bottles to resuspend the extract. The extract was then filtered through Whatman filter paper #1 (Whatman, Maidstone, UK), and then the methanol was removed by evaporation in a fume hood for 24 h at room temperature. The caper extract was next centrifuged for 10 min at 15 °C, 4000 RCF to remove the remaining particulate matter, followed by storage at −20 °C. On the day of electrophysiological recording, we thawed the caper extract and diluted it 1:100 in bath solution (see below) immediately before use.

**Channel subunit cRNA preparation and Xenopus laevis oocyte injection**. As previously described[7], we generated cRNA transcripts encoding human KCNE1, KCNE3, KCNQ1, KCNQ2, KCNQ3, KCNQ4 or KCNQ5 by in vitro transcription using the mMessage mMachine kit (Thermo Fisher Scientific), after vector linearization, from cDNA sub-cloned into plasmids incorporating Xenopus laevis β-globin 5′ and 3′ UTRs flanking the coding region to enhance translation and cRNA stability. We generated mutant KCNQ1, KCNQ2 and KCNQ3 cDNAs by site-directed mutagenesis with a QuikChange kit (Stratagene, San Diego, CA) and prepared the cRNAs as above. We injected defolliculated stage V and VI Xenopus laevis oocytes (Xenoocyte, Dexter, MI, US and NASCO, Fort Atkinson, WI, US) with KCNE and/or KCNQ cRNAs (2–20 ng). We incubated the oocytes at 16 °C in ND96 oocyte storage solution containing penicillin and streptomycin, with daily washing, for 2–4 days prior to TEVC recording.

**Two-electrode voltage clamp**. We performed TEVC at room temperature using an OC-725C amplifier (Warner Instruments, Hamden, CT) and pClamp10 software (Molecular Devices, Sunnyvale, CA) 2–5 days after cRNA injection as described in the section above. For recording, we placed the oocytes in a small-volume oocyte bath (Warner) and viewed them with a dissection microscope. We sourced chemicals from Sigma. We studied effects of 1% caper extract and of compounds previously identified in caper and/or cilantro extract, solubilized directly in bath solution (in mM): 96 NaCl, 4 KCl, 1 MgCl$_2$, 1 CaCl$_2$, 10 HEPES (pH 7.6). To minimize conversion of rutin to quercetin we made working solutions up on the day of experimentation and shielded the solution from light before and during experiments. We introduced extract or compounds into the oocyte recording bath by gravity perfusion at a constant flow of 1 ml per minute for 3 min prior to recording. Pipettes were of 1–2 MΩ resistance when filled with 3 M KCl.

We recorded currents in response to voltage pulses between −120 or −80 mV and +40 mV at 10 mV intervals from a holding potential of −80 mV, to yield current-voltage relationships and examine activation and inactivation. We also used other voltage protocols as shown in the figures to further investigate inactivation. We analyzed data using Clampfit (Molecular Devices) and Graphpad Prism software (GraphPad, San Diego, CA, USA); values are stated as mean ± SEM. We plotted raw or normalized tail currents versus prepulse voltage and fitted with a single Boltzmann function:

Eq. 1

$$g = \frac{(A_1 - A_2)}{\left\{1 + exp[V_{\frac{1}{2}} - V/Vs]\right\}y + A_2}$$

where $g$ is the normalized tail conductance, $A_1$ is the initial value at $-\infty$, $A_2$ is the final value at $+\infty$, $V_{1/2}$ is the half-maximal voltage of activation and $V_s$ the slope factor. We fitted activation and deactivation kinetics with single exponential functions.

**Chemical structures and silico docking**. We plotted and viewed chemical structures and electrostatic surface potential using Jmol, an open-source Java viewer for chemical structures in 3D: http://jmol.org/. For in silico ligand docking predictions of binding to KCNQ1, KCNQ1/KCNE1 and KCNQ1/KCNE3, we performed unguided docking of quercetin and (E)-2-dodecanal to predict potential binding sites, using SwissDock with CHARMM forcefields[58,59] and closed and open state models of either channel previously developed by others[36,49] and the cryo-electron microscopy-derived human KCNQ1-calmodulin and KCNQ1-PIP$_2$-calmodulin/KCNE3 structures[37].

**Statistics and reproducibility**. All values are expressed as mean ± SEM. One-way ANOVA was applied for all tests; all $p$ values were two-sided.

**Reporting summary**. Further information on research design is available in the Nature Research Reporting Summary linked to this article.

## Data availability

All datasets and materials are available upon reasonable request. Source data used to generate the charts present in the paper is available as Supplementary Data 1 and 2.

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

## Acknowledgements

We are grateful to Angele De Silva (University of California, Irvine) and Dr. Gianina Panaghie (Weill Cornell Medical College) for generating mutant channel constructs, and to Bo Abbott for the image of pickled capers. *Capparis spinosa* bush photo credit: Lazaregagnidze (Wikipedia). G.W.A. is indebted to Helen M. Abbott for many years of horticultural and culinary advice. This study was supported by the National Institutes of Health, National Institute of General Medical Sciences and National Institute of Neurological Disorders and Stroke (GM130377 and NS107671 to G.W.A.).

## Author contributions

K.E.R. prepared caper extracts, performed the oocyte experiments and analyses, prepared most of the figure panels, and edited the paper. G.W.A. performed in silico structural analyses, wrote the paper and helped prepare the figures.

## Competing interests

The authors declare no competing interests.
