## [Peer Review File · Communications Biology]

Reviewers' comments:

Reviewer #1 (Remarks to the Author):

The manuscript by Redford and Abbott focus on the effect of the flavonoid Quercetin on the activity of voltage-gated channels encoded by the KCNQ gene family. Using the *Xenopus* oocytes expression system, Redford and Abbott found that Quercetin can hyperpolarize oocytes expressing KCNQ1, KCNQ2 and KCNQ3 channels.

For this work, Quercetin was extracted from pickled capers. Both crude extract and purified Quercetin were able to hyperpolarize oocytes expressing KCNQ channels. Intriguingly, the mechanism of action seems to differ for different KCNQ channels. In the case of KCNQ2 and KCNQ3, Quercetin induced a modest increase in current amplitude. In contrast, for the heteromeric KCNQ2/KCNQ3, Quercetin produced instead a shift towards more negative potential in the voltage dependence of activation. For KCNQ1, Quercetin boosted the kinetic of activation as well as shifting the voltage-dependence to more negative values. In addition, Quercetin also enhanced increased inactivation. The authors argue that the apparent increase in activation seems to be due to an increase in the activation rate. Nonetheless, the effect on the activation seemed to prevail over the enhanced inactivation, producing a net hyperpolarization of oocytes expressing KCNQ1.

The mechanism of action of Quercetin remains to be identified. Yet, the author made strides toward identifying key residues involved in the interaction between KCNQ channels and Quercetin. Combining *in silico* docking and experimental mutagenesis, the authors identified residues in the VSD and pore domain of KCNQ channels that were able to alter or mitigate the effect of Quercetin on channel activity. The authors mentioned in the manuscript that Quercetin seems to have multiple binding sites in KCNQ1. *In silico* and experimental data support this idea. However, the high concentration of Quercetin used for the assays (100 μ M) constitute a concern to be considered when drawing this conclusion.

The manuscript is well-written and the data provided by the authors seems to support their conclusions. However, I have two concerns to be addressed before this manuscript can be published.

Concerns:

1. In pages 8 and 9, the authors indicate the deactivation is made slower by the action of Quercetin. However, the channels are also brought faster into an inactivated state. This seems to indicate that deactivation is driven from the an "stabilized" inactivated state in the presence of the flavonoid, making it slower. So, it is possible that recovering from inactivation and not deactivation *per se* could be determining the overall kinetic of deactivation. The authors should elaborate on this.

2. On page 9, the authors stated "no hook apparent in tail current and no visible pre-pulse inactivation" when referring to their results with the mutant KCNQ1-R231A. Isn't this result expected from a channel that is open at all tested potential. In other words, isn't it fair to presume that the channels can be already "inactivated"? Please clarify the need for this statement.

Reviewer #2 (Remarks to the Author):

Paper Review „The ubiquitous flavonoid quercetin is an atypical KCNQ potassium channel activator“

In their manuscript "The ubiquitous flavonoid quercetin is an atypical KCNQ potassium channel activator" Redford and Abbott examine the effect of quercetin and its glycosylated form rutin on

different KCNQ channels in presence and absence of the auxiliary subunit KCNE. The results are detailed and the performed electrophysiological experiments are well conducted. However, the performed *in silico* binding site prediction and experimental evaluation of interaction is not convincing. The implementation of binding site scores and the comparison of the results with known channel ligands could provide the reader a better understanding of the findings. Mutagenesis scanning is highly suggested to evaluate docking. Allosteric interactions can be tested via MD simulations if they cannot be addressed experimentally. Furthermore, I would like to encourage the authors to address the following major/minor points:

Major Points

1. The authors mentioned the extraction procedure of pickled capers. However, composition of extract is not clear and can be heavily influenced by even small changes of the procedure. It is necessary to quantify the final concentration of quercetin in the tested extract. Technically, this detection/quantification may be done via HPLC or mass spec..

In line with the quantity check the quality of compounds used have to be controlled. On page 14 the authors state that the pickling process can promote the conversion of rutin to quercetin. I totally agree with this. In addition other handling processes can promote this conversion as well. The authors should explain how this chemical conversion was controlled/avoided by compound handling and during their experiments.

2. Quercetin is a well-known flavonoid that is often discussed in different aspects of pharmacology and toxicology. The authors should give a short overview about the current data on toxicity aspects of the compound. This point is highly relevant because a use of a natural compound with multiple side effects at a very high dose of 100 μM limits its value as a drug candidate and as a lead structure for drug development. As a tool compound it still might have some value if no better tools are available.

3. Page 5, line 6: increase of inactivation for KCNQ1 can be seen under the influence of extract. The figure as presented does not convince me of a marked effect on inactivation. Possibly, quantification of the effect or a different depiction of the data given in figure 1H would be helpful.

4. Figure 2A-B: merged picture for quercetin is not necessary. Structural and electrostatical informations of quercetin and rutin would be easier to compare, if both structures would have a similar orientation. Please correct.

5. Page 5 line 19: the inhibition of KCNQ2/3 channels by rutin was declared as moderate. However, effect in figure 2D seems to be not significantly different from control. Please check statistical significance of tail current differences.

6. Page 6, line 7: Please add the values for $\Delta V_{0.5\text{activation}} \pm \text{SEM}$ for homomeric KCNQ2 and KCNQ3* channels.

7. Figure 2A and Figure 3D are identical. Please remove one of them.

8. Page 8, last sentence: The authors declare that quercetin stabilizes the open state and destabilizes the closed state. Markov modelling would help to mathematically test this hypothesis. The authors may consider applying this technique.

9. Page 9: The *in silico* docking lead to multiple potential binding sites, which is a common problem with binding site prediction, especially for very low-affinity compounds. One possible binding site located at the extracellular surface of the VSD was evaluated by mutation of R228 and R231. However, these residues seem not to interact directly with quercetin by ionic/hydrophobic interactions

since side chain of these residues are orientated away from the binding site. The authors may consider assaying direct interactions by classical mutational scanning of residues that are predicted to interact directly with quercetin by hydrophobic, aromatic or ionic interactions. Indirect interactions may be tested by careful MD simulations.

10. Page 13 / Discussion: The authors should focus on their findings. I would encourage the authors to short passages describing traditional use in different cultures and the possible chances for further development of drugs against different diseases. As the compounds effects are very mild at very high doses the compounds value may be as tool compound but the clinical/pharmacological use is clearly overstated.

Minor Points

Page 3, line 8: 6 Rename: Transmembrane segments (S1-S6) instead of (S).

Page 3, line 13: Please give a reference for the paragraph.

Figure Legend 1A: Explanation for pH is missing. > pore helix.

Figure 1H, I: It might be of help to quantify the effect of inactivation by mathematical re-analysis of recorded traces.

Figure 3E-F: Visibility of retigabine binding site and discussed amino acids is limited. A close-up view would be instrumental.

Figure 4D/E: Please add a capture for quercetin.

Reviewer #3 (Remarks to the Author):

In the paper, Redford KE and Professor Abbott identified that quercetin, the most commonly consumed flavonoid prepared from caper extract, is a KCNQ activator with an atypical selectivity and mechanism. Quercetin activated KCNQ1, KCNQ1/KCNE1, KCNQ2/Q3 and KCNQ4, but not KCNQ5 channels heterologously expressed in oocyte. Combining molecular docking, mutagenesis and electrophysiological recording, several residues located at the external section of the voltage sensor and the pore region were thought to be important for quercetin activity. In addition, the synergic effects of quercetin and other natural compounds were also examined. These data supported that quercetin potentiates KCNQ channels in an atypical manner and suggested that the modulation may contribute to the medicinal effects produced by quercetin. In general, the work was well organized and conducted. However, several concerns should be addressed before publishing.

1. The activation effects of quercetin on KCNQ1, KCNQ2/KCNQ3 was only measured at 100 μ M, a dose-dependence is suggested.
2. Quercetin potentiates KCNQ1/KCNE1 complex, whereas inhibits KCNQ1 currents at high stimulus potential. Due to the potential pharmacological activity, the effects of quercetin on native IKs current and action potential duration (APD) of cardio myocyte are helpful.
3. The channel names in figures are inconsistent. (KCNQ1, Q1-R243A, Q1-F340W/E1, KCNQ1-R231A/E3, KCNQ1/E3, KCNQ1/KCNE3)

My co-author and I were very pleased by the positive response to our manuscript and we thank the reviewers for their careful consideration and constructive critiques. We have responded to the specific questions with additional analysis of electrophysiological data, new docking studies using the now-available cryoEM-resolved structures for KCNQ1 and KCNQ1-KCNE3, new analysis of docking results, and editing of the figures and text. We describe the changes point-by-point below.

Reviewer #1 (Remarks to the Author):

The manuscript is well-written and the data provided by the authors seems to support their conclusions. However, I have two concerns to be addressed before this manuscript can be published.

>We greatly appreciate the highly positive review.

Concerns:

1. In pages 8 and 9, the authors indicate the deactivation is made slower by the action of Quercetin. However, the channels are also brought faster into an inactivated state. This seems to indicate that deactivation is driven from the an “stabilized” inactivated state in the presence of the flavonoid, making it slower. So, it is possible that recovering from inactivation and not deactivation per se could be determining the overall kinetic of deactivation. The authors should elaborate on this.

>This is a great point and we now discuss this (last sentence of page 8)

2. On page 9, the authors stated “no hook apparent in tail current and no visible pre-pulse inactivation” when referring to their results with the mutant KCNQ1-R231A. Isn't this result expected from a channel that is open at all tested potential. In other word, isn't it fair to presume that the channels can be already “inactivated”? Please clarify the need for this statement.

>It is possible they are already inactivated, although in a previous paper (Panaghie et al., 2008 Biophysical Journal) we showed that in F340W-IKs channels, which exhibit constitutive activation but voltage-dependent inactivation, the R231A mutation removes inactivation. We have expanded the sentence to discuss this (last sentence of page 9).

Reviewer #2 (Remarks to the Author):

Paper Review „The ubiquitous flavonoid quercetin is an atypical KCNQ potassium channel activator”

In their manuscript “The ubiquitous flavonoid quercetin is an atypical KCNQ potassium channel activator” Redford and Abbott examine the effect of quercetin and its glycosylated form rutin on different KCNQ channels in presence and absence of the auxiliary subunit KCNE. The results are detailed and the performed electrophysiological experiments are well conducted. However, the performed in silico binding site prediction and experimental evaluation of interaction is not convincing. The implementation of binding site scores and the comparison of the results with known channel ligands could provide the reader a better understanding of the findings. Mutagenesis scanning is highly suggested to evaluate docking. Allosteric interactions can be tested via MD simulations if they cannot be

addressed experimentally. Furthermore, I would like to encourage the authors to address the following major/minor points:

>We greatly appreciate the detailed and thoughtful review.

Major Points

1. The authors mentioned the extraction procedure of pickled capers. However, composition of extract is not clear and can be heavily influenced by even small changes of the procedure. It is necessary to quantify the final concentration of quercetin in the tested extract. Technically, this detection/quantification may be done via HPLC or mass spec..

In line with the quantity check the quality of compounds used have to be controlled. On page 14 the authors state that the pickling process can promote the conversion of rutin to quercetin. I totally agree with this. In addition other handling processes can promote this conversion as well. The authors should explain how this chemical conversion was controlled/avoided by compound handling and during their experiments.

>To minimize conversion of rutin to quercetin we made working solutions up on the day of experimentation and shielded the solution from light before and during experiments. We must have been at least successful enough using this approach within the resolution of our system because, e.g., we found no effects of rutin on KCNQ1 but strong effects of quercetin (Figure 4A-C). If we had a significant degree of conversion, we would have observed effects with the "rutin" solution. We now add this explanation in the Discussion (page 14) and in the Methods (page 20). We now quote USDA values for quercetin content of pickled versus raw capers in the Discussion (page 14) as we are not sure when we will have access to mass spec facilities.

2. Quercetin is a well-known flavonoid that is often discussed in different aspects of pharmacology and toxicology. The authors should give a short overview about the current data on toxicity aspects of the compound. This point is highly relevant because a use of a natural compound with multiple side effects at a very high dose of 100 μ M limits its value as a drug candidate and as a lead structure for drug development. As a tool compound it still might have some value if no better tools are available.

>In a 2007 critical review in Food and Chemical Toxicology (Harwood et al., 2007) it was concluded that it is safe to add quercetin to food. We have added this reference. However, it is true that 100 μ M is a relatively high dose. We see effects at lower doses (see Figure 10a) but we agree that potency is a limiting factor and that quercetin is more likely useful as a tool compound and perhaps a lead compound for further medicinal chemistry. We have added this to the Discussion (page 18).

3. Page 5, line 6: increase of inactivation for KCNQ1 can be seen under the influence of extract. The figure as presented does not convince me of a marked effect on inactivation. Possibly, quantification of the effect or a different depiction of the data given in figure 1H would be helpful.

>We have quantified the hook in response to this comment, showing that quercetin increases the hook by 45% (new Figure 1K, L).

4. Figure 2A-B: merged picture for quercetin is not necessary. Structural and electrostatical informations of quercetin and rutin would be easier to compare, if both structures would have a similar orientation. Please correct.

>We have corrected the orientation and removed the merged picture. We also circled the quercetin moiety within rutin to improve clarity.

5. Page 5 line 19: the inhibition of KCNQ2/3 channels by rutin was declared as moderate. However, effect in figure 2D seems to be not significantly different from control. Please check statistical significance of tail current differences.

>The P value was 0.32, we have edited the sentence to reflect this.

6. Page 6, line 7: Please add the values for $\Delta V_{0.5}$ activation +/- SEM for homomeric KCNQ2 and KCNQ3* channels.

>We have added the values as requested.

7. Figure 2A and Figure 3D are identical. Please remove one of them.

>We removed Figure 3D.

8. Page 8, last sentence: The authors declare that quercetin stabilizes the open state and destabilizes the closed state. Markov modelling would help to mathematically test this hypothesis. The authors may consider applying this technique.

>This is a really good suggestion. We have started a longer-term project to record single channel activity of KCNQ2/3 channels in the absence and presence of small molecule modulators such as quercetin, GABA, followed by Markov modeling. This will take the best part of a year to do properly even when we can access the lab again and so we feel this should be outside the scope of the current project. In the absence of this we have softened the language about open and closed states.

9. Page 9: The in silico docking lead to multiple potential binding sites, which is a common problem with binding site prediction, especially for very low-affinity compounds. One possible binding site located at the extracellular surface of the VSD was evaluated by mutation of R228 and R231. However, these residues seem not to interact directly with quercetin by ionic/hydrophobic interactions since side chain of these residues are orientated away from the binding site. The authors may consider assaying direct interactions by classical mutational scanning of residues that are predicted to interact directly with quercetin by hydrophobic, aromatic or ionic interactions. Indirect interactions may be tested by careful MD simulations.

>We performed additional docking studies using the new cryo-EM-derived structures from the MacKinnon lab of KCNQ1-calmodulin and KCNQ1-PIP₂-KCNE3-calmodulin (the docking studies in the original manuscript, which we have left in for comparison, are from open- and closed-state structural models by the Sanders group). In the new docking studies, the highest-ranked docking clusters in terms of most negative delta G values were for quercetin poses in which quercetin was predicted to H-bond to the R231 sidechain. We have now added these data to Figure 6 (panels E, F) and Figure 9 (panel F) and describe them in the text. We believe this adds further valuable support for R231 being an important binding site for quercetin, reinforced also by our mutagenesis results (last sets of panels in Figures 6 and 9).

10. Page 13 / Discussion: The authors should focus on their findings. I would encourage the authors to short passages describing traditional use in different cultures and the possible chances for further development of drugs against different diseases. As the compounds effects are very mild at very high doses the compounds value may be as tool compound but the clinical/pharmacological use is clearly overstated.

>We have shortened as requested, removing 2 pages of text.

Minor Points

Page 3, line 8: 6 Rename: Transmembrane segments (S1-S6) instead of (S).

>Corrected as requested.

Page 3, line 13: Please give a reference for the paragraph.

>Corrected as requested.

Figure Legend 1A: Explanation for pH is missing. > pore helix.

>Corrected as requested.

Figure 1H, I: It might be of help to quantify the effect of inactivation by mathematical re-analysis of recorded traces.

>We have added this (Figure 1K, L).

Figure 3E-F: Visibility of retigabine binding site and discussed amino acids is limited. A close-up view would be instrumental.

>Added as requested.

Figure 4D/E: Please add a capture for quercetin.

>Added as requested.

Reviewer #3 (Remarks to the Author):

In the paper, Redford KE and Professor Abbott identified that quercetin, the most commonly consumed flavonoid prepared from caper extract, is a KCNQ activator with an atypical selectivity and mechanism. Quercetin activated KCNQ1, KCNQ1/KCNE1, KCNQ2/Q3 and KCNQ4, but not KCNQ5 channels heterologously expressed in oocyte. Combining molecular docking, mutagenesis and electrophysiological recording, several residues located at the external section of the voltage sensor and the pore region were thought to be important for quercetin activity. In addition, the synergic effects of

quercetin and other natural compounds were also examined. These data supported that quercetin potentiates KCNQ channels in an atypical manner and suggested that the modulation may contribute to the medicinal effects produced by quercetin. In general, the work was well organized and conducted. However, several concerns should be addressed before publishing.

>We greatly appreciate the highly positive review.

1. The activation effects of quercetin on KCNQ1, KCNQ2/KCNQ3 was only measured at 100 μ M, a dose-dependence is suggested.

>We have a dose response in the final Figure (10a), which may have made it difficult to find. We put it at the end to allow comparison between channels without duplicating results.

2. Quercetin potentiates KCNQ1/KCNE1 complex, whereas inhibits KCNQ1 currents at high stimulus potential. Due to the potential pharmacological activity, the effects of quercetin on native IKs current and action potential duration (APD) of cardio myocyte are helpful.

>We agree that this will be a fascinating experiment both ex vivo and in vivo in the appropriate animal model. We intend to study this in the future but feel it is outside the scope of the current project.

3. The channel names in figures are inconsistent. (KCNQ1, Q1-R243A, Q1-F340W/E1, KCNQ1-R231A/E3, KCNQ1/E3, KCNQ1/KCNE3)

>Apologies – we have cleaned this up in the revised manuscript.

REVIEWERS' COMMENTS:

Reviewer #1 (Remarks to the Author):

The authors has properly addressed my concerns

Reviewer #2 (Remarks to the Author):

My concerns and suggestions have been addressed convincingly.

Reviewer #3 (Remarks to the Author):

No further comments. All concerns by the reviewer have been addressed.